# Enhanced Electroluminescence from a Silicon Nanocrystal/Silicon Carbide Multilayer Light-Emitting Diode

**DOI:** 10.3390/nano13061109

**Published:** 2023-03-20

**Authors:** Teng Sun, Dongke Li, Jiaming Chen, Yuhao Wang, Junnan Han, Ting Zhu, Wei Li, Jun Xu, Kunji Chen

**Affiliations:** 1School of Electrical Science and Engineering, Collaborative Innovation Centre of Advanced Microstructures, Jiangsu Provincial Key Laboratory of Advanced Photonic and Electrical Materials, Nanjing University, Nanjing 210000, China; 2ZJU-Hangzhou Global Scientific and Technological Innovation Centre, School of Materials Science and Engineering, Zhejiang University, Hangzhou 311200, China

**Keywords:** Si nanocrystals, SiC, phosphorous, LED

## Abstract

Developing high-performance Si-based light-emitting devices is the key step to realizing all-Si-based optical telecommunication. Usually, silica (SiO_2_) as the host matrix is used to passivate silicon nanocrystals, and a strong quantum confinement effect can be observed due to the large band offset between Si and SiO_2_ (~8.9 eV). Here, for further development of device properties, we fabricate Si nanocrystals (NCs)/SiC multilayers and study the changes in photoelectric properties of the LEDs induced by P dopants. PL peaks centered at 500 nm, 650 nm and 800 nm can be detected, which are attributed to surface states between SiC and Si NCs, amorphous SiC and Si NCs, respectively. PL intensities are first enhanced and then decreased after introducing P dopants. It is believed that the enhancement is due to passivation of the Si dangling bonds at the surface of Si NCs, while the suppression is ascribed to enhanced Auger recombination and new defects induced by excessive P dopants. Un-doped and P-doped LEDs based on Si NCs/SiC multilayers are fabricated and the performance is enhanced greatly after doping. As fitted, emission peaks near 500 nm and 750 nm can be detected. The current density-voltage properties indicate that the carrier transport process is dominated by FN tunneling mechanisms, while the linear relationship between the integrated EL intensity and injection current illustrates that the EL mechanism is attributed to recombination of electron–hole pairs at Si NCs induced by bipolar injection. After doping, the integrated EL intensities are enhanced by about an order of magnitude, indicating that EQE is greatly improved.

## 1. Introduction

In order to meet the growing demand of data-carrying capacity, optical telecommunication has attracted much attention [1,2,3]. In this regard, developing efficient Si-based light-emitting devices (LED) is the key issue to be addressed. Silicon nanocrystals (Si NCs) are believed to be the most promising option to realize this due to their novel physical properties [4,5]. Photoluminescence (PL) spectral shift and enhanced quantum efficiency were observed in Si NCs with various dot sizes, while the stable quantum yield of Si NCs with polymer can reach 60–70% [5,6,7,8]. Recently, Si NCs-based LEDs with different emitting wavelengths have been achieved [9,10,11,12]. Further research is still necessary to explore the potential of Si NCs-based LEDs.

Doping intentionally in a semiconductor is a key method for enhancement of electrical and optical properties. Aside from the great improvement in conductivity, phosphorous (P) and boron (B) dopants can change the electronic structures of Si NCs, thus affecting the optical properties [13,14,15,16,17]. More interestingly, P dopants will introduce a deep level in 2 nm sized Si NCs and emit near-infrared light near 1200 nm [18,19,20]. In addition, carrier tunneling between Si NCs is dependent on the barrier of the host matrix in stacked structures [21]. Silicon carbide (SiC) as a host matrix is a promising candidate due to a narrower and modulative bandgap [22,23,24]. 

In our previous work, we investigated electron spin resonance of size-varied Si NCs/SiC multilayers induced by P dopants and carrier transport behaviors of various P-doped Si NCs/SiC multilayers, respectively [16,24]. Electroluminescence (EL) of Si NCs/SiC multilayers and Si NCs embedded in a SiC matrix have also been discussed in detail [25,26,27]. In the present work, 4 nm sized Si NCs/SiC multilayers with various P-doping ratios are fabricated. PL intensities are enhanced first and then decreased after gradually introducing P dopants. A similar tendency happens to Hall mobility. As fitted, PL peaks at 500 nm, 650 nm and 800 nm are observed, which are ascribed to amorphous SiC (650 nm), Si NCs (800 nm) and the surface states between SiC and Si NCs (500 nm), respectively. The enhanced PL intensities are caused by passivation of Si dangling bonds at the surface of Si NCs by P dopants. The quenched PL intensities are ascribed to enhanced carriers-induced Auger recombination and the emerging defects caused by excessive P dopants. We also fabricate un-doped and P-doped LEDs with the structure of aluminum (Al)/4 nm sized Si NCs/SiC multilayers/indium-tin oxide (ITO). The EL intensities of both the un-doped and P-doped LEDs are enhanced after gradually increasing the applied current. The fitted EL spectra have emission peaks near 500 nm and 750 nm, which are similar to the PL spectra. It is found that the radiative recombination in Si NCs accounts for the majority (above 90%) of emissions, manifesting that it is easier for electrons and holes being recombined radiatively in Si NCs. Only part of the radiative recombination induced by quantum-confined Si NCs occurs in the surface states between Si NCs and SiC during the tunneling process. After doping, not only are the electrical properties of the devices greatly improved but the EL intensities at the same applied current are also enhanced by about an order of magnitude, indicating an order of magnitude increase in external quantum efficiency (EQE).

## 2. Experiment

*a*-Si/SiC multilayers are deposited on p-Si ((1 0 0), 0.01 Ω·cm) wafer and quartz substrates in plasma-enhanced chemical vapor deposition (PECVD) system. The radio frequency, RF power and chamber temperature are kept at 13.56 MHz, 5 W and 250 °C, respectively. As for *a*-Si/SiC multilayers, the gas mixtures of phosphine (PH_3_) gas (5%, H_2_ dilution) and silane (SiH_4_) gas are introduced into the chamber to deposit the P-doped a-Si sublayers. Various doping levels are achieved by changing nominal gas ratio between silane and phosphine ([PH_3_]/[SiH_4_]). The gas flow of SiH_4_ during the Si sublayer deposition process is fixed at 10 sccm, while the flow of PH_3_ is varied from 0, 2, 5 and 10 sccm. Gas mixtures of CH_4_ (50 sccm) and SiH_4_ (5 sccm) are introduced to deposit SiC sublayers. The as-deposited samples are annealed at 450 °C for 1 h under nitrogen (N_2_) ambient for dehydrogenation and then thermally annealed at 1000 °C for 1 h under nitrogen (N_2_) ambient to form Si NCs/SiC multilayers and annealed-SiC layers. Then, Al electrode is deposited at the back of p-Si substrate through magnetron sputtering under room temperature and the samples are annealed at 420 °C for 0.5 h under N_2_ ambient to reduce the dangling bonds and heat defects in the films and also to promote ohmic contact between the film and the electrodes. At last, ITO electrode is deposited also by magnetron sputtering under a shadow mask and the active area was about 0.03 cm^2^. The temperature of the ITO deposition process is kept at 200 °C.

Images of Si NCs/SiC multilayers are measured by transmission electron microscopy (TEM, TECNAI G^2^F20 FEI, Hillsboro, TX, USA), while the TEM samples are fabricated by focused ion beam scanning electron microscopy (FIB-SEM) system. Hall mobility of Si NCs with various doping ratios is detected by van der Pauw (VDP) geometry (LakeShore 8400 series, Lorain, OH, USA) at room temperature. Steady-state PL spectra equipped with 325 nm continuous He-Cd laser by Edinburgh FLS 980 spectrophotometer at room temperature. Fourier transform infrared (FTIR, Thermo Scientific Nicolet iS20, Waltham, MA, USA) spectra are measured to analyze chemical bond composition of samples. Absorption spectra are measured at room temperature by Shimadzu UV-3600 spectrophotometer (Shimadzu Co., Kyoto, Japan). Finally, the EL measurements of the Si NCs/SiC multilayers were carried out at room temperature with Edinburgh FLS 980 spectrophotometer by applying positive bias to the Al back electrode, and ITO is used as the negative electrode. The PL and EL spectra are all corrected by the fundamental correction file of the instrument because the detected spectra are wide. The original PL of Si NCs/SiC multilayers with various doping ratios, EL spectra of un-doped LED and EL spectra of 1% P-doped LED are shown in Appendix A, respectively. In addition, the photometer used in this work can only cover the visible range well; the EL emissions in near-infrared (NIR) range require further research.

## 3. Results and Discussion

The cross-sectional TEM images of Si NCs/SiC multilayers are shown in Figure 1a,b. Periodic structures and the abrupt interface between Si and SiC sublayers can be obviously observed in Figure 1a. On top of the Si NCs/SiC multilayers is platinum (Pt), which is used to protect the TEM samples from being etched by gallium (Ga) ions during the focused ion beam process. As measured, the thicknesses of the Si and SiC sublayers are ~4.0 nm and ~2.0 nm, respectively. The high-resolution TEM image of a single Si NC in the Si sublayer is also shown in the inset of Figure 1a, manifesting that the ~4 nm sized Si NCs are formed in Si sublayers after a high-temperature annealing process (1000 °C, N_2_ ambient). The lattice distance is ~0.314 nm, which is corresponding to the lattice distances of Si (1 1 1). Si NCs with well constrained dot sizes are formed in Si sublayers. Meanwhile, Si NCs are close to each other and separated by amorphous structure according to Figure 1b, indicating good size-controllability of Si NCs in the stacked structures [25].

Figure 2a shows PL intensities tendency of Si NCs/SiC multilayers with various P-doping ratios excited by the 325 nm laser. It is found that PL intensities are first enhanced and then decreased after gradually introducing P dopants into our samples. The maximum PL intensities are achieved when the P-doping ratio is 1%. Meanwhile, Hall mobility, as measured, increases first and then decreases with increasing P-doping ratios, and a maximum Hall mobility of 1.74 cm^2^/V·s is achieved at 2.5% P-doped samples according to Figure 2b. As noted, there exist considerable defects, such as Si dangling bonds at the surface of Si NCs, which will trap free carriers and suppress recombination of electrons and holes. Interestingly, P dopants can passivate the dangling bonds at the surface of Si NCs, leading to luminescence being improved. Excessive P dopants, however, will introduce amounts of free carriers and do damage to the lattice, which will enhance Auger recombination and introduce new defects, respectively [18,28,29]. Similar doping properties are also found in Si NCs/SiC systems, and P dopants will introduce further defects/Si vacancies in an ultra-small Si NCs/SiC system [16]. The enhanced PL intensity is, thus, attributed to passivation of Si dangling bonds by P dopants. Although the highest mobility is achieved when the P-doping ratio is 2.5%, P dopants will provide considerable free carriers, leading to stronger Auger recombinations, which will quench the PL of samples [30,31]. When doping ratio continues to increase, Hall mobility is decreased, manifesting that enhanced impurity scattering and lattice damage may occur, further suppressing luminescence. After fitting the PL spectrum of the 1% P-doped samples, PL peaks at ~500 nm (~2.5 eV), ~650 nm (~1.9 eV) and ~800 nm (1.6 eV) can fit the spectrum well. Inspired by the effective mass approximation (EMA) model, the theoretical bandgap of 4 nm sized Si NCs is ~1.7 eV(~750 nm) and similar EL spectra originated from 4 nm sized Si NCs were also observed in our previous work [27,32,33,34,35]. We attribute the PL peak near 800 nm to the 4 nm sized Si NCs in the Si NCs/SiC system. 

Here, we also deposit SiC single layers. The annealing process of the amorphous SiC single layers is in consistence with that of Si NCs/SiC multilayers. We first measure the PL spectra of un-annealed SiC (R = 10). Pronounced PL peaks near 650 nm can be detected according to Figure 3. As reported, 1200 °C or 1250 °C is believed to be the crystallization temperature for SiC, and there is no crystallization after 100 h at 1000 °C in dry air [36,37]. Our previous work suggests that there is also no crystallization for SiC in Si NCs/SiC multilayers through the same annealing process, as measured by X-ray diffraction [24]. Chen et al. have reported similar results in amorphous SiC films, and the emissions can be enhanced via nitrogen doping [38]. Based on our existing knowledge, we attribute the PL peak near 650 nm to amorphous SiC. Annealed SiC with various R ([CH_4_]/[SiH_4_]), where the gas flow of SiH_4_ is fixed at 5 sccm, have also been fabricated. In Appendix A, PL peaks around 500 nm can be detected in all the annealed SiC single layers with various R. The maximum PL intensities of SiC single layers are reached when R is 12. Appendix A exhibits the absorption spectra of annealed SiC single layers with various R. It is found that the absorption spectra change slightly with R. Guided by Tauc’s plot, we use the relationship of ${\left(\alpha h\upsilon \right)}^{2}\propto (h\upsilon -E_{g})$ to simulate the absorption edge of the annealed SiC single layers. As provided in the inset of Appendix A, the estimated optical bandgap of SiC also changes slightly with R and is ~2.5 eV (~500 nm) [39]. In order to study the chemical composition of the annealed SiC with various R, the FTIR spectra of the un-annealed and annealed SiC single layers with various R are measured in Appendix A, respectively. Comparing such spectra before and after annealing, we can find that the function group bond of Si–H near 2090 cm^−1^ disappears after thermally annealing regardless of R. It demonstrates that the thermal annealing process excludes hydrogen atoms completely. Signals of the functional group bond of Si–C near the wavenumber of 790 cm^−1^ can be detected in the samples with R = 10, 12 and 14. The highest intensities of the functional group bond of Si–C are reached when R is 12. Meanwhile, all the samples show the functional group bond of Si–O near the wavenumber of 1110 cm^−1^ and have similar intensities of signals. As for samples with R = 12 and 14, the functional group bond of C=O near the wavenumber of 2400 cm^−1^ can be detected, indicating that such samples may be C-rich SiC layers [40,41,42]. The PL emissions near 500 nm may be related to the functional group bond of Si–C. In addition, similar PL peaks near 500 nm have been found in annealed amorphous SiC alloys in our previous work [43]. It is also found that pronounced peaks near 500 nm can be observed in SiC_x_, SiC_x_O_y_ nanoclusters and silicon-rich silicon oxide enriched with C [44,45,46]. They attribute such PL or EL peaks to the surface states of SiC nanoclusters. As for the annealed SiC single layers, Si NCs may be formed in the SiC dielectric, although the annealing temperature (1000 °C) is not enough to form SiC particles. Si NCs embedded in a SiC matrix have been widely investigated in our previous work [25,26,27]. Consequently, we attribute the PL peaks near 500 nm in the Si NCs/SiC multilayers to the surface states between amorphous SiC and Si NCs. 

Figure 4a shows the schematic representation of the Si NCs/SiC multilayer LED prepared in this work, the structure of which is ITO/Si NCs/SiC multilayers/p-Si/Al. At the back of the p-Si substrate is an Al electrode, which is connected to the positive terminal of a power meter. On top of the p-Si substrate are Si NCs/SiC multilayers and then the circle ITO electrode, which is connected to the negative terminal of the power meter. The area of the active circular regions upon the samples is 0.03 cm^2^. Further, 1% P-doped Si NCs/SiC multilayers-based LEDs are selected due to the tendency of PL intensities and P doping behaviors. Figure 4b displays injection current density as a function of applied voltage (*J-V*) curves of Si NCs/SiC multilayers before and after doping. The current density increases slowly at first and then increases dramatically when the applied voltage is high enough in both un-doped and doped LEDs. It is also found that the electrical properties of the Si NCs/SiC multilayer LED are greatly improved by P dopants, i.e., on-set voltage (improved to ~5.7 V) and conductivity. In order to better understand the vertical carrier transport mechanisms of our devices, we estimate the quantity *J*/*V*^2^ as a function of *1*/*V* in Figure 4c. According to the inset of Figure 4c, we can see that a straight line can fit the data at high applied voltage well, indicating that carrier transport is dominated by a Fowler–Nordheim (FN) tunneling mechanism when applied voltage is high [27,47].

Furthermore, EL spectra of 0% and 1% P-doped LEDs with various applied currents are measured, as shown in Figure 5a,b, respectively. EL intensities are gradually enhanced with increasing applied current both in un-doped and P-doped LED. To better understand the EL emissions of the LEDs, we select the EL spectrum of 1% P-doped LED with 13 mA applied current to fit. It is found that EL peaks near 500 nm and 750 nm can fit the spectrum well according to Figure 5c. As for EL peaks near 500 nm, we attribute it to the surface states between SiC and Si NCs, which has been discussed above. Interestingly, such EL peaks near 500 nm cannot be observed in 8 nm sized Si NCs/SiC multilayer LED devices regardless of doping. Appendix A shows the EL spectrum of 1% P-doped 8 nm sized Si NCs/SiC multilayers-based LED at the applied current of 10 mA. The EL emissions, accordingly, from the surface states between SiC and Si NCs (500 nm) may be attributed to quantum-confined ultra-small Si NCs [48,49]. Further, the main EL peaks near 750 nm account for the majority of the EL spectrum. Similar results were observed in Si NCs/SiC multilayers crystallized by a KrF pulsed excimer laser in our previous work [35]. In addition, the peaks near 750 nm are consistent with the bandgap of 4 nm sized Si NCs, which has been estimated before [27,50]. The peak near 750 nm, consequently, is ascribed to the 4 nm sized Si NCs. As shown in the inset of Figure 5c, the bright light, which can be seen by naked eyes when our LED works, indicates that our P-doped LEDs have better performance than in our previous work. Figure 5d shows the main luminescence center (above 90%) emissions of Si NCs (750 nm). The proportion of the integrated EL peak near 750 nm is decreased slightly, while that of the integrated EL peak near 500 nm is increased slightly. Electrons and holes can be injected by FN tunneling into 4 nm sized Si NCs and then radiative recombination occurs, leading to EL peaks near 750 nm being emitted from the LED. With applied voltage increasing, more electrons can tunnel through SiC, making radiative recombination of the surface states between SiC and Si NCs easier to occur. 

Figure 6 exhibits the energy diagram of the LED. The energy of ITO, Al, p-Si and the bandgap of Si NCs are taken from the previous work [27,35,51,52]. The energy bands of Al and ITO are −4.1 eV and −4.7 eV, respectively. The energy of conduction band and valence band of p-Si substrate are −4.1 eV and −5.2 eV, respectively. As measured before and extracted by PL and EL spectra, the bandgap of 4 nm sized Si NCs is 1.7 eV. According to the following relation (1) and (2):(1)$$E_{g}=3\Delta CB+1.1eV$$
(2)ΔVB=2ΔCB
where $E_{g}$, ΔCB and ΔVB are the bandgap of Si NCs, the energy shift of conduction band and the energy shift of valence band, respectively, the estimated energy shifts of conduction band and valence band are 0.2 eV and 0.4 eV, respectively [10,53]. The energy of conduction band and valence band of Si NCs, thus, are −3.9 eV and −5.6 eV, respectively. The barrier height between Si and SiC dielectric matrix has been estimated in our previous work: V_0e_ (0.4 eV) and V_0h_ (0.8 eV) for electron and hole, respectively [27]. The energy of conduction band and valence band of SiC are −3.5 eV and −6.4 eV, respectively. Figure 6 also displays the schematic of EL recombination mechanisms in our samples. With applied voltage, large amounts of electrons and holes are injected into Si NCs by FN tunneling. Most of the electrons and holes are recombined radiatively and emit the EL peak near 750 nm (~1.7 eV). During the tunneling process, some of the electrons are recombined radiatively in SiC dielectric and emit an EL peak near 500 nm (~2.5 eV).

As noted, FN tunneling will occur with a voltage as low as 5.7 V (the on-set voltage for 1% P-doped LED), which is consistent with the turn-on voltage when EL signals are observed. It should be noted that injection current will increase with increasing applied voltage accordingly. A linear relationship between integrated EL intensity and applied current can be observed both in the un-doped and P-doped LEDs according to Figure 7a. It indicates that bipolar injection of electrons and holes rather than impact ionization occurs in our devices regardless of doping [27,54]. Thus, it is believed that the EL emissions originated from recombination of bipolar injected electrons and holes through FN tunneling, as shown in Figure 6. By comparing the integrated EL intensities under the same applied current between 0% and 1% P-doped devices in Figure 7b, the EL intensities in 1% P-doped devices are ~eight times those in 0% P-doped devices. Meanwhile, the EQE of the P-doped devices is improved by ~eight times 0% P-doped devices. It is illustrated that P dopants can not only greatly improve LED performance, including conductive properties and on-set voltage, but also EL emissions due to the doping effect of P dopants, namely providing considerable free carriers and improving surface structures of Si NCs.

## 4. Conclusions

In conclusion, 4 nm sized Si NCs/SiC multilayers are fabricated in this work. PL peaks originated from 4 nm sized Si NCs (800 nm), amorphous SiC (620 nm) and surface states between Si NCs and SiC (500 nm) can be observed. PL intensities can be enhanced after P doping due to passivation of Si dangling bonds by P dopants and quenched when P dopants are excessive. Bright LEDs with the structure of ITO/4 nm sized Si NCs/SiC multilayers/p-Si substrate/Al are finally fabricated. The EL band can be fitted by 500 nm peak and 750 nm peak, which are attributed to surface states between Si NCs and SiC and Si NCs, respectively. It is indicated that the carrier transport process is dominated by an FN tunneling mechanism. EL mainly comes from recombination between electrons and holes in Si NCs due to bipolar injection. After doping, performance of devices is greatly improved. Aside from conductivity and on-set voltage, integrated EL intensities under the same applied current are enhanced by about an order of magnitude, indicating the EQE of Si NCs/SiC multilayer LED can be improved greatly by doping. It is demonstrated that P-doped Si NCs/SiC multilayers is a promising method to realize more efficient Si-NCs-based LED devices and Si-based optical telecommunication.

## Figures and Tables

**Figure 1 nanomaterials-13-01109-f001:**
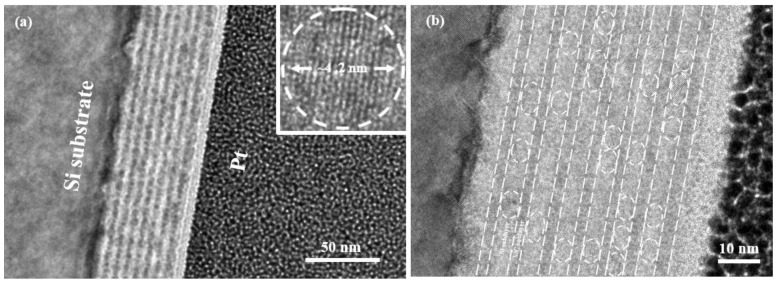
Cross-sectional TEM images of Si NCs/SiC multilayers at 50 nm scale (**a**) and 10 nm scale (**b**). Inset is the high-resolution TEM images of a single Si NC.

**Figure 2 nanomaterials-13-01109-f002:**
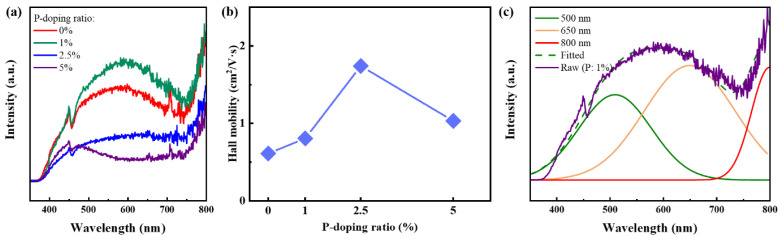
(**a**) PL spectra of Si NCs/SiC multilayers with various P-doping ratios; (**b**) Hall mobility of Si NCs/SiC multilayers with various P-doping ratios; (**c**) fitted PL spectra of 1% P-doped Si NCs/SiC multilayers.

**Figure 3 nanomaterials-13-01109-f003:**
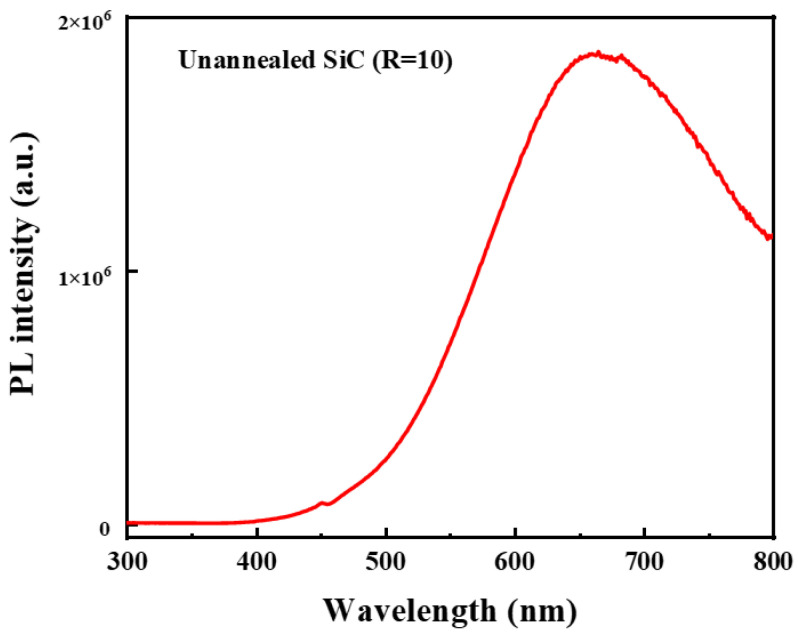
PL spectrum of the un-annealed SiC (R = 10).

**Figure 4 nanomaterials-13-01109-f004:**
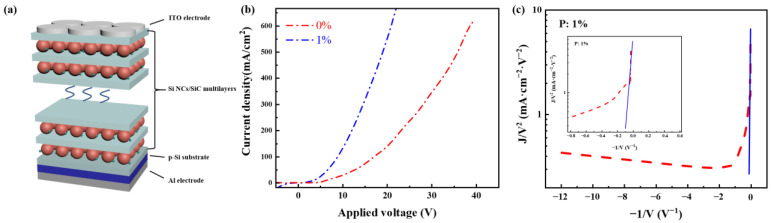
(**a**) Schematic structure of the Si NCs/SiC multilayer LED fabricated in this work; (**b**) current density of 0% and 1% P-doped LEDs as a function of applied voltage; (**c**) FN tunneling plot of 1% P-doped data; inset is the zoom of the region fitting the FN mechanism.

**Figure 5 nanomaterials-13-01109-f005:**
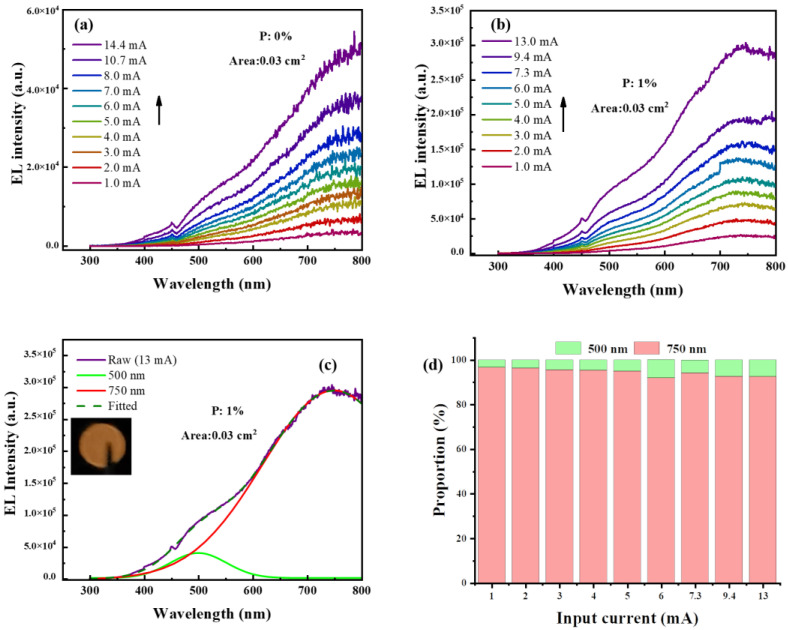
EL spectra of 0% (**a**) and 1% (**b**) P-doped Si NCs/SiC multilayer LEDs with various applied currents; (**c**) fitted spectra of 1% P-doped LED with 13 mA applied current. Inset is the photograph of the EL emissions from the LED; (**d**) proportion of different EL peaks with various applied currents.

**Figure 6 nanomaterials-13-01109-f006:**
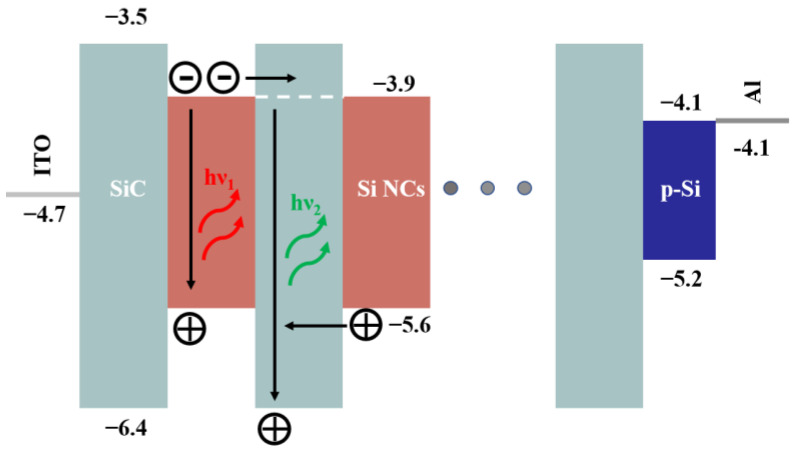
Schematic of EL recombination mechanisms and energy diagrams of the Si NCs/SiC multilayer LED.

**Figure 7 nanomaterials-13-01109-f007:**
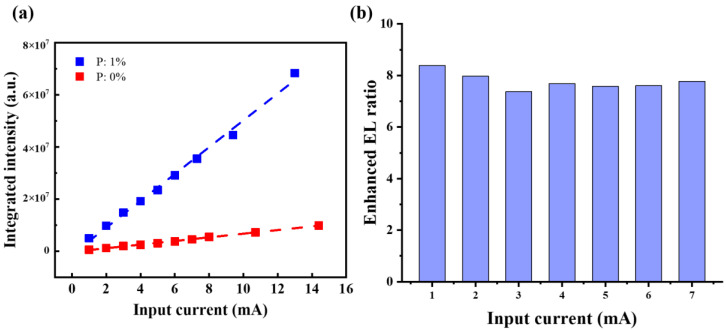
(**a**) Integrated intensity of EL spectra of 0% and 1% P-doped LEDs with various applied currents; (**b**) enhanced EL ratio between 0% and 1% P-doped LEDs with various applied currents.

## Data Availability

Data will be made available on request.

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
