# Peer review of "Enhanced Electroluminescence from a Silicon Nanocrystal/Silicon Carbide Multilayer Light-Emitting Diode"

_nanomaterials, 2023, doi:10.3390/nano13061109_

Round 1

Reviewer 2 Report

The paper is interesting, but it needs to be improved. I attached my comments.

Round 2

Reviewer 2 Report

The authors answered to my comments. However, in the revised manuscript they inserted new PL and EL figures in which the tendency of peaks changes (Si NC emission relative to SiC/Si NC surface states), in my opinion the authors have to argument this major change to reviewers and editor.

The EL spectra in manuscript-v1 extended until 900 nm, in the new manuscript is only up to 800 nm. Why?

Please provide a zoom of the region fitting the FN mechanism in the I-V curve.

Round 3

Reviewer 2 Report

The authors answered to my comments. The paper can be accepted for publication.

Author Response

Thank you for your comments!